# Avidity observed between a bivalent inhibitor and an enzyme monomer with a single active site

Shiran Lacham-Hartman[1], Yulia Shmidov[2], Evette S. Radisky[3], Ronit Bitton[2], David B. Lukatsky[4]*, Niv Papo[1]*

**1** Avram and Stella Goldstein-Goren Department of Biotechnology Engineering and the National Institute of Biotechnology in the Negev, Ben-Gurion University of the Negev, Beer-Sheva, Israel, **2** Deprtment of Chemical Engineering and the Ilse Katz Institute for Nanoscale Science & Technology, Ben-Gurion University of the Negev, Beer-Sheva, Israel, **3** Department of Cancer Biology, Mayo Clinic Comprehensive Cancer Center, Jacksonville, Florida, United States of America, **4** Department of Chemistry, Ben-Gurion University of the Negev, Beer-Sheva, Israel

* papo@bgu.ac.il (NP); lukatsky@bgu.ac.il (DBL)

**Data Availability Statement:** All relevant data are within the manuscript and its Supporting information files.

## Abstract

Although myriad protein–protein interactions in nature use polyvalent binding, in which multiple ligands on one entity bind to multiple receptors on another, to date an affinity advantage of polyvalent binding has been demonstrated experimentally only in cases where the target receptor molecules are clustered prior to complex formation. Here, we demonstrate cooperativity in binding affinity (*i.e.*, avidity) for a protein complex in which an engineered dimer of the amyloid precursor protein inhibitor (APPI), possessing two fully functional inhibitory loops, interacts with mesotrypsin, a soluble monomeric protein that does not self-associate or cluster spontaneously. We found that each inhibitory loop of the purified APPI homodimer was over three-fold more potent than the corresponding loop in the monovalent APPI inhibitor. This observation is consistent with a suggested mechanism whereby the two APPI loops in the homodimer simultaneously and reversibly bind two corresponding mesotrypsin monomers to mediate mesotrypsin dimerization. We propose a simple model for such dimerization that quantitatively explains the observed cooperativity in binding affinity. Binding cooperativity in this system reveals that the valency of ligands may affect avidity in protein–protein interactions including those of targets that are not surface-anchored and do not self-associate spontaneously. In this scenario, avidity may be explained by the enhanced concentration of ligand binding sites in proximity to the monomeric target, which may favor rebinding of the multiple ligand binding sites with the receptor molecules upon dissociation of the protein complex.

## Introduction

Numerous biological activities in nature rely on polyvalent interactions, in which multiple ligands on one entity transiently [1] or irreversibly [2, 3] bind to multiple targets (e.g.,

**Funding:** This work was supported by the Worldwide Cancer Research (grant number 20-0238) and the European Research Council Proof of Concept grant (grant number 875197) to N.P., and the United States - Israel Binational Science Foundation (grant number 2019303) to N.P and E. S.R, and the Israel Science Foundation (ISF) (grant number 1004/20) to D.B.L. The funders had no role in study design, data collection and analysis, decision to publish, or preparation of the manuscript.

**Competing interests:** The authors have declared that no competing interests exist.

receptors) on another [4, 5]. These biological activities include nutrient transport (e.g., hemoglobin), immune recognition (e.g., antibodies), and signal transduction (e.g., receptor tyrosine kinases) [5, 6]. Each non-covalent binding interaction can be characterized by its binding affinity, with the accumulated strength of many individual affinities referred to as functional affinity, or avidity.

The literature classifies polyvalent interactions according to four main groups based on the type of molecular entities that are involved: (i) cooperative interactions following protein aggregation, (ii) intermolecular (allosteric) cooperative interactions, such as binding of hemoglobin to iron atoms, (iii) intramolecular cooperative interactions, such as chelating effects (i.e., in which both the ligand and the receptor are multivalent) and (iv) interannular cooperativity describing the interplay of multiple intramolecular binding interactions [7–10]. An avidity effect, on the other hand, refers to cooperativity that results from chelating effects [7, 11]. Although the majority of polyvalent interactions are collectively much stronger than their corresponding monovalent interactions (*i.e.*, exhibit stronger avidities [5, 9]), there are also examples for negative cooperativity in polyvalent binding due to destabilization of one ligand unit by another [12]. In addition, polyvalency can imbue a ligand with novel properties upon binding to a receptor target, such as the ability to cluster [13, 14] or to disengage from surface-anchored receptors [15]. These properties allow polyvalent ligands to either agonize or antagonize biological interactions that are crucial for intra- and extracellular structural, metabolic, signaling, and regulatory pathways [16].

These agonistic and antagonistic effects have been elucidated from systems in which ligands interact with surface-anchored receptors. Examples include transcription factor binding to multiple DNA regions [17], and influenza virus fusion with the sialic acids found at the host cell surface via the former's hemagglutinin (HA) glycoproteins and the latter's cell adhesion mediation properties [18]. Additional examples include conjugating engineered polypeptides that contain multiple integrin-binding RGD motifs spaced by SGSGSGSG linkers to a surface for cell adhesion [19], attachment of immune cells to epithelial cells via glycoproteins to generate an immune response [20], and others (reviewed in [5] and [16]).

Polyvalent interactions may also impact the binding selectivity of a ligand to its target by inducing steric stabilization of specific interactions and/or by enhancing the local concentration of a ligand [21]. The size and distribution of polyvalent ligands in the vicinity of their targets may reduce the interaction of the target with other natural ligands [22]. For example, Lees *et al.*, [23] have shown that a polymer containing multiple sialic acids was able to inhibit the adhesion of influenza virus to erythrocytes by binding to multiple hemagglutinin (HA) proteins on the virus surface. Moreover, the interaction of this multivalent polymer with multiple HA proteins also prevented binding of specific antibodies to HA due to steric hindrance imposed by the multivalency during binding. Polyvalent ligands may also increase the concentration of ligand binding sites in the proximity of their target receptor, such that when the ligand–receptor protein complex dissociates, the receptor target can re-associate with multiple ligand binding sites such that a polyvalent system favors rebinding with the ligand [22, 24].

In theory these advantages may also confer avidity effects between polyvalent ligands and naturally monomeric, soluble receptors which are incapable of self-association except if mediated by the scaffolding of the ligand. Engineering of such polyvalent ligands may provide a novel strategy to enhance both affinity and specificity of protein-protein interactions. However, the affinity implications of polyvalency in such systems has not been explored, possibly because of a lack of suitable and easily accessible model systems [21]. Therefore, we aimed to close this gap by investigating the effect of avidity on a system in which all the binding sites in the polyvalent protein ligand are fully functional (*i.e.*, no destabilization in the polyvalent ligand) and the target protein, in its soluble form, is monomeric and may not self-associate or

cluster. The complex formed between the serine protease mesotrypsin and its cognate inhibitor, the human amyloid precursor protein inhibitor (APPI), from the Kunitz domain family, represents an excellent system for that purpose.

As a result of specific evolutionary mutations, mesotrypsin exhibits a distinctive resistance to almost all biological inhibitors including those of the Kunitz domain family [25–29]. Indeed, compared with other trypsin isoforms, the binding affinities of mesotrypsin with human Kunitz domains are 2–4 orders of magnitude weaker [30]. These low affinities (which are expressed as high inhibition constants ($K_i$)) render mesotrypsin a good soluble target for exploration of potential avidity effects, since improvements in the relatively weak binding affinities should be easily detectable.

APPI, which is abundant in nature, is composed of 58-amino acids, with a MW of approximately 6 kDa [31]. The canonical loop within the APPI scaffold serves as a recognition site for mesotrypsin [30, 32], to which it binds as a competitive inhibitor with 1:1 stoichiometric ratio [33–35] and inhibition constant of about 133 nM [33, 36]. Here, we envisioned that the APPI Kunitz domain could be arranged in tandem repeats to generate a polyvalent scaffold, inspired by the natural arrangement of tandem Kunitz domains in bikunin, another Kunitz domain family member. In bikunin, the two nonidentical Kunitz domains occur sequentially in a single protein chain, with each subunit possessing a distinct inhibitory spectrum; the N-terminal domain targets elastases, while the C-terminal domain targets trypsin-like serine proteases [37].

Here, we generated a stable homodimeric APPI in which both canonical inhibitory loops, one on each monomer, are correctly folded and fully functional. We show that the inhibitory activity of each loop in polyvalent APPI is over three-fold more potent than that of the corresponding loop in monovalent APPI. Using mesotrypsin, which does not self-associate spontaneously, we probed how each inhibitory loop in the dimer could bind each mesotrypsin monomer and thus provide support for mesotrypsin dimerization upon binding to the bivalent APPI. We also propose a simple kinetic model for mesotrypsin—APPI homodimer binding, quantitatively explaining the experimentally observed cooperativity in binding affinity (*i.e.*, avidity). In so doing, we provide a framework for showing that the valency of ligands may affect avidity in protein–protein interactions that include targets that are not surface-anchored and do not self-associate spontaneously.

## Results and discussion

Polyvalent interactions play an important role in many biochemical processes [5]. The conjugation of multiple copies of a ligand to a scaffold facilitates the simultaneous interaction of the latter with multiple target molecules, which strengthens binding affinity through an avidity effect [22, 38, 39]. A key criterion for avidity is prior clustering of the target molecules, such as occurs when receptors are immobilized on a cell surface or solid support antecedent to ligand binding [5, 40]. This effect has been found mostly in antibody–antigen interactions [1] in which the antigen is membrane-anchored. To the best of our knowledge, no attempts have been made to investigate the presence of avidity effects in soluble protein complexes in which a polyvalent ligand interacts with target molecules that are not in close proximity to each other prior to polyvalent protein complex formation. In such instances, it may be less trivial to achieve an avidity effect, as avidity requires that the unanchored, unclustered and freely diffusing monomeric target molecules (which have the potential to interfere with multiple binding to the ligand) exhibit superior binding affinity to polyvalent ligands than to monovalent ligands. The current research was designed to develop and explore avidity under such conditions, that is, when there is no clustering of the target molecules prior to their binding to the polyvalent ligand.

## Generating the dimer construct gene in a *Pichia pastoris* expression system

To examine the feasibility of inducing binding avidity in a soluble protein complex comprised of a polyvalent ligand that binds freely diffusing, unclustered, monomeric target molecules, we chose the APPI–mesotrypsin complex, which was first characterized in Salameh *et al.* [33], as proof of concept. To satisfy the first criterion, namely, that each unit of the polyvalent ligand be fully functional, it was necessary to generate a correctly folded recombinant APPI dimer having two fully functional inhibitory units. Additionally, properties such as ligand density (spacing) and orientation were considered in selecting an appropriate linker. Many studies have suggested that the linker can play a crucial role in facilitating ligand binding and that it can be optimized to allow correct protein folding or to obtain optimal biological activity in fusion proteins [19, 22, 41].

The mesotrypsin-APPI heterodimer is shaped like a mushroom, where mesotrypsin is the mushroom cap and APPI is the elongated mushroom stem [32]. Both the N- and C-termini of APPI are located opposite its inhibitory loop, so that by connecting two APPI molecules in tandem, we connect two mushroom-shaped complexes at the base of their stems via the linker. Even a short linker would allow >50 Å between the two mesotrypsin molecules due to the length of the two stems, resulting in little risk of steric hindrance between mesotrypsin sub-units. Our larger concern was thus achieving efficient independent folding and correct disul-fide bond configuration within each of the two APPI domains, which we hoped to maximize by selecting a well-characterized flexible linker composed of three repetitions of GlyGlyGly-GlySer (designated GGGGS×3), of ~48 Å in length [41].

The gene construct for dimeric APPI was generated by PCR assembly (**S1A Fig in** S1 File) and transformed into *P. pastoris*. We then extracted the genomic DNA and amplified the tar-get genes with AOX primers. The PCR products were separated by electrophoresis on 1% aga-rose gel and the bands representing the amplified APPI genes were confirmed, as shown in **S1B Fig in** S1 File. The sequence of the APPI dimer gene construct containing the GGGGS×3 linker was also verified (**S1C Fig in** S1 File).

## Large-scale protein production and purification of APPI variants

The recombinant APPI variants were purified using nickel-affinity chromatography, which is possible because of their C-terminal His tags (Fig 1A and 1B). Size exclusion chromatography (SEC) was then used to eliminate impurities, such as imidazole residuals, from the APPI monomer or dimer (Fig 1C and 1D). Subsequently, we analyzed the purified proteins on 15% SDS-PAGE followed by InstantBlue Coomassie staining to confirm the size of the proteins (Fig 1E and 1F and S1A Raw images).

As expected (given a MW of approximately 9 kDa for the APPI monomer containing a His tag and restriction enzyme site [31]) dimeric APPI showed a predominant band at MW 15 kDa (Fig 1F). However, a significant and unexpected difference was observed between APPI dimer concentrations obtained from analytical and functional assays. Specifically, whole pro-tein concentrations obtained from the absorbance spectrum of the APPI dimer at 280 nm were significantly higher than the inhibitory unit concentrations obtained from its inhibitory activity in the presence of bovine trypsin (i.e., 268 μM and 38 μM from absorbance at 280 nm and active sites titration, respectively). From these observations, we inferred the presence of several recombinant dimeric APPI conformations, of which only some displayed the correctly folded and fully functional canonical loops (*i.e.*, enzyme recognition sites) that serve as inhibi-tory units in their interaction with trypsins (*i.e.*, bovine trypsin and mesotrypsin). On the basis of the monomeric structure of the APPI Kunitz domain, which contains three intramolecular disulfide bridges, the monomeric units that constitute the APPI dimer may potentially include

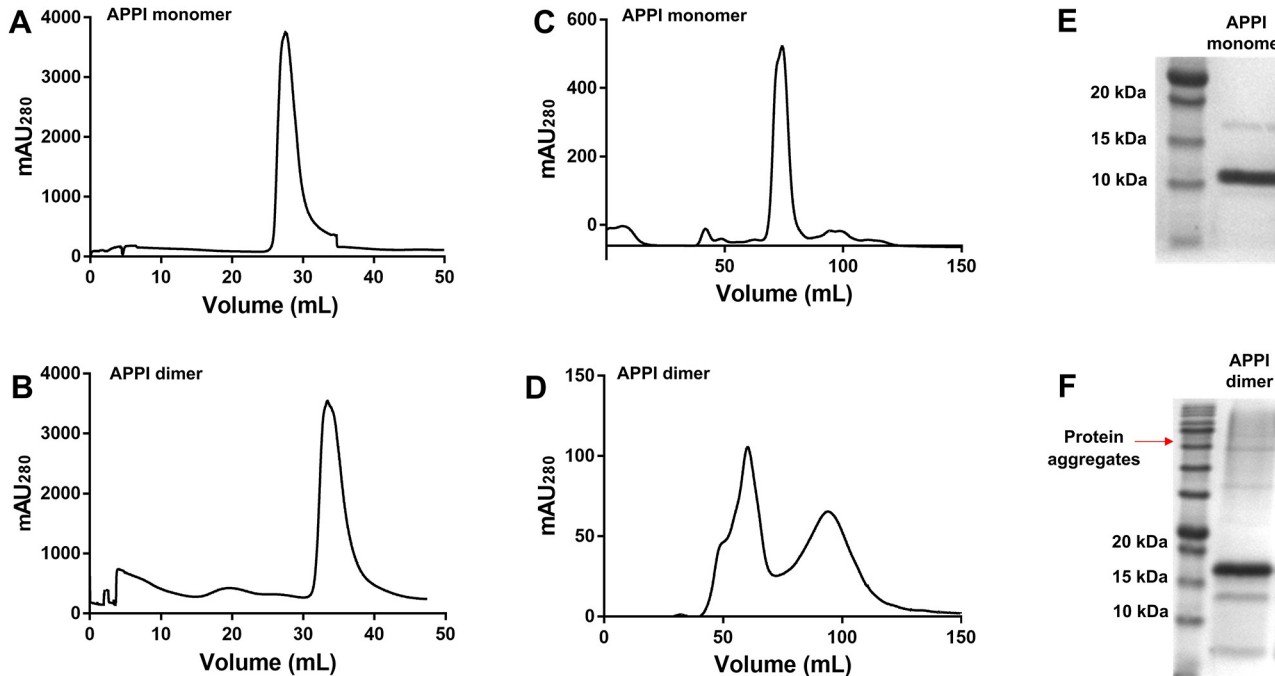

**Fig 1. Purification of APPI variants.** Nickel affinity chromatography (**A** and **B**) and size-exclusion chromatography (SEC) (**C** and **D**) of recombinant amyloid precursor protein inhibitor (APPI) monomers (**A** and **C**) and dimers (**B** and **D**). The left peak in (**D**) is dimeric APPI, whereas the right peak is imidazole, which was used for protein elution in the nickel column. For all chromatograms, the X-axis represents the elution volume and the Y-axis represents absorbance at 280 nm. 15% SDS-PAGE analysis of the SEC-purified APPI variants (**E** and **F**).

non-natural, mixed pairs of disulfide bonds that enable only partial protein folding and that are incompatible with correct folding of the inhibitory loop. During protein folding, the correct formation of intramolecular disulfide bonds within each monomeric unit of the APPI dimer is essential for the activity of the inhibitory loop [42]. It is particularly crucial in the context of enzyme–inhibitor complexes, such as trypsin–APPI, in which the inhibitory loop of APPI binds to the active site of mesotrypsin in a site specific manner, as per the lock-and-key interaction model [43].

To generate an APPI dimer in which both the displayed canonical inhibitory loops were fully functional, we applied an additional protein purification step involving affinity chromatography with a bovine trypsin column (Fig 2A) to isolate the fraction of dimeric APPI that is correctly (and fully) folded. Two separate peaks were observed and analyzed by 15% SDS-PAGE (Fig 2B). The protein fraction represented by Peak 2 exhibited superior inhibition of mesotrypsin than the protein fraction represented by Peak 1 (Peak 1 showed a large difference between two concentration values, the first obtained from active sites titration (i.e., 11 μM) and the second that was calculated according to absorbance at 280 nm (i.e., 50 μM). We then compared the dimeric APPI fractions corresponding to affinity chromatography Peaks 1 and 2 with monomeric APPI by means of an additional 15% SDS-PAGE analysis (Fig 2C and S1B Raw images), and these fractions were then used for further functional characterization of the APPI protein dimer. Next, we determined the monomeric and dimeric APPI protein concentrations using absorbance at 280 nm (with extinction coefficients of 13,325 and 22,180 $M^{-1}$ $cm^{-1}$, respectively). Approximately 20 mg and 0.4 mg of purified APPI monomer and dimer were produced from a 1 L yeast culture, respectively. Finally, MALDI-TOF spectra

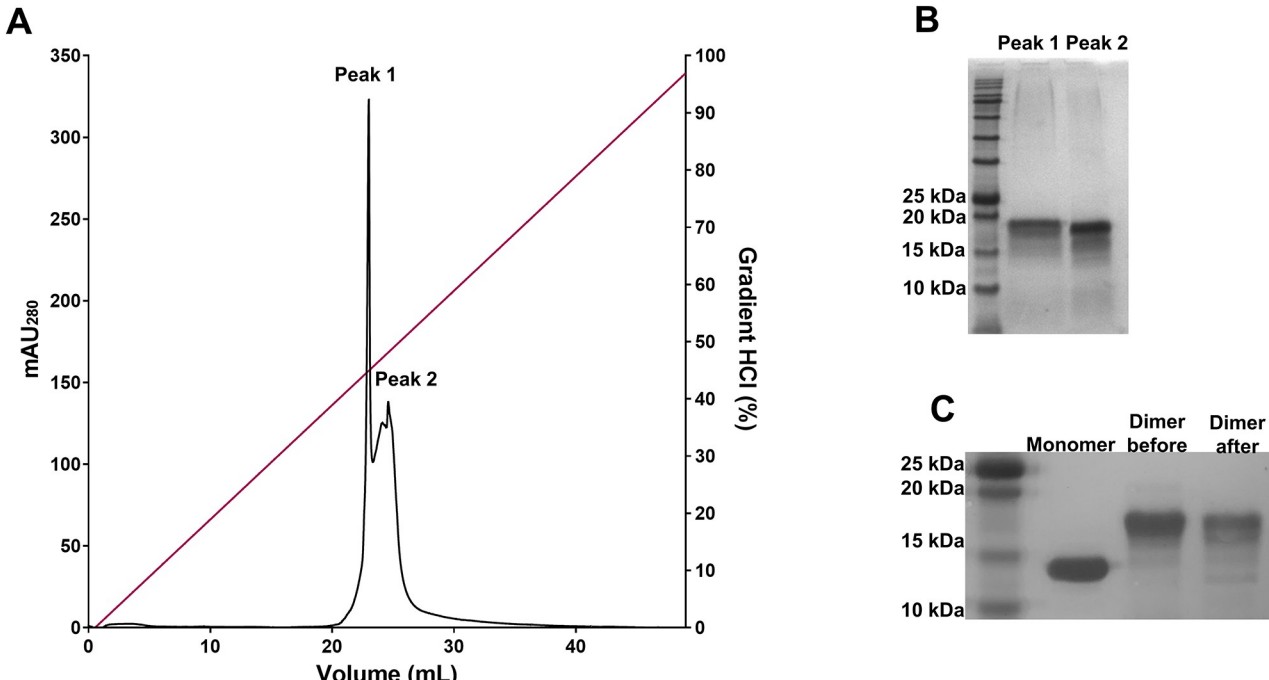

**Fig 2. Final purification step to obtain fully functional dimeric APPI.** Using an affinity chromatography approach, dimeric APPI was eluted from a bovine trypsin column using a 0–100 mM HCl gradient (**A**). The protein fraction eluted in Peak 2 was chosen for further functional (protease inhibition) experiments, because it exhibited greater inhibition potency toward trypsin in comparison with the protein fraction eluted in Peak 1. 15% SDS-PAGE analysis of the purified dimeric APPI fractions (**B**) and comparison of purified monomeric APPI with dimeric APPI before and after purification (**C**).

confirmed the MW of the purified monomeric and dimeric APPI variants (**S2A and S2B Fig in** S1 File).

## APPI dimer titration analysis is consistent with 1:2 APPI:trypsin binding stoichiometry

Monomeric APPI is a 1:1 tight-binding picomolar inhibitor of trypsin, which can be used as a titrant to determine concentrations of APPI preparations [44]. To determine whether APPI dimer can simultaneously engage two molecules of trypsin, the APPI inhibitory unit concentration (in the APPI monomer and dimer) was determined by trypsin titration analysis and compared with concentrations determined by absorbance (**S1 Table in** S1 File). To determine concentrations by titration assay (Fig 3), we used a known quantity of bovine trypsin, which has known molecular mass, and our two inhibitors (APPI monomer and dimer), which have known molecular masses. A comparison between (a) the concentration of active inhibitory units determined from the titration (Fig 3) and (b) the inhibitor molecule concentrations measured by absorbance at 280 nm showed that nearly double the amount of trypsin was quenched as could be explained by a 1:1 binding model (**S1 Table in** S1 File). However, when the concentration of APPI dimer molecules was calculated from the titration results assuming 1:2 APPI:trypsin binding stoichiometry, the concentrations determined by titration and absorbance were highly consistent. These data provide evidence that APPI dimer can bind simultaneously to two molecules of trypsin. Because of the structural similarity of trypsin family members, we anticipate that the APPI dimer is also capable of binding simultaneously to two molecules of mesotrypsin.

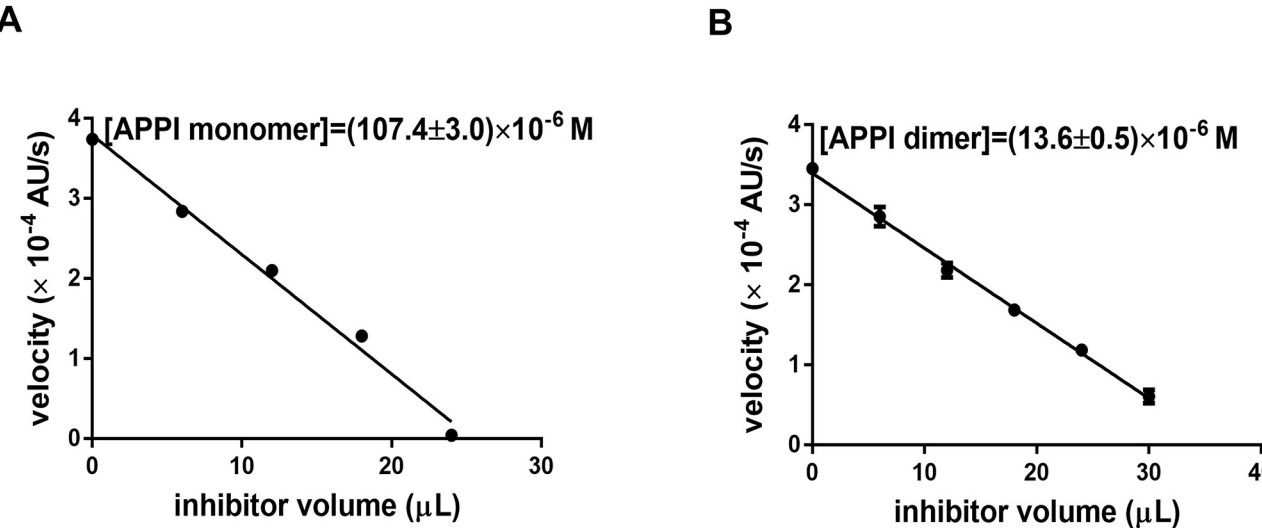

**Fig 3. Titration of inhibition units.** Bovine trypsin titration curves of monomeric (**A**) and dimeric (**B**) APPI. The X-axis represents the APPI volumes added to the reaction and the Y-axis represents the reaction velocity (*i.e.*, velocity of substrate cleavage by bovine trypsin) at several APPI dilutions. The experiment setup is described in the Materials and methods section.

### The mesotrypsin affinity of a single inhibitory unit is enhanced in dimeric compared with monomeric APPI

To assess the binding of monomeric and dimeric APPI to mesotrypsin, we carried out enzyme inhibition experiments that monitored cleavage of the chromogenic substrate Z-GPR-pNA by mesotrypsin in the presence of varying APPI concentrations. First, we determined the Michaelis-Menten constant ($K_m$) of z-GPR-pNA binding to mesotrypsin. As shown in Fig 4A, the $K_m$ constant was calculated using nonlinear regression fitting to the Michaelis-Menten equation. The obtained value was 25.1±2.4 μM, consistently with previous studies [36]. Next, we

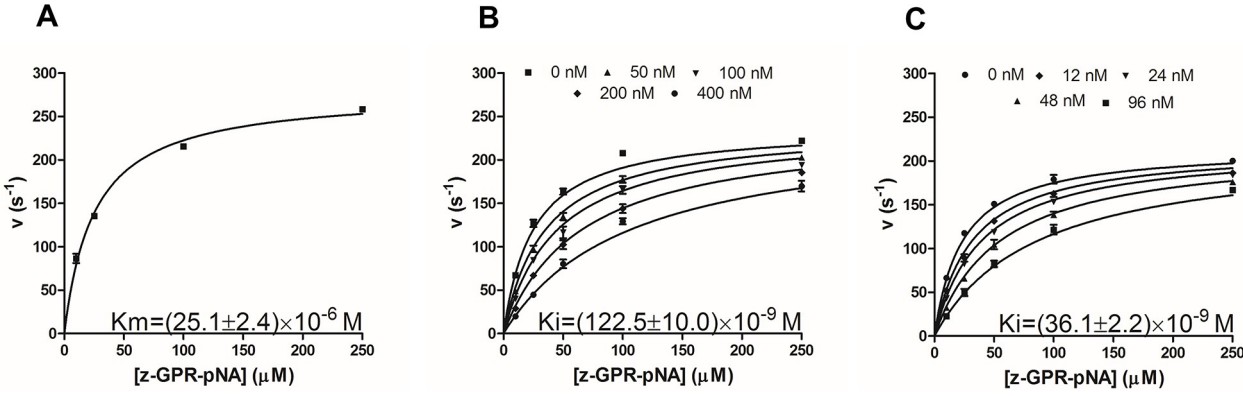

**Fig 4. Determination of inhibition constants.** Determination of the Michaelis-Menten constant ($K_m$) for binding of the chromogenic substrate Z-GPR-pNA to mesotrypsin (**A**). Kinetics of mesotrypsin inhibition by the APPI monomer (**B**) and dimer (**C**). In all experiments, substrate concentration was 5–250 μM and enzyme concentration was 0.25 nM. The concentration of the APPI inhibitor unit was 0–400 nM for the monomer and 0–96 nM for the dimer (double the concentration of the APPI dimer molecule possessing two inhibitor units). Reactions were followed spectroscopically for 5 min, and initial rates were determined from the increase in absorbance caused by the release of p-nitroaniline (at 410 nm). Data were globally fitted by multiple regression to the classic competitive inhibition equation (Eq 20; see Materials and methods). Raw data can be found in SI.

determined the affinity of the inhibitory units in monomeric and dimeric APPI proteins for mesotrypsin using a competitive inhibition model to calculate the inhibition constant, for consistency with prior work [33, 36]. For the dimer, this analysis assumes an absence of cooperativity between the first and second enzyme molecule binding events, and thus yields a single $K_i$ value that applies to each inhibitory unit of the dimer. We observed a classic competition pattern of inhibition for both inhibitors (Fig 4B and 4C). We found that the apparent inhibition constant of a single inhibitory unit in the APPI dimer was $K_i$ = 36.1±2.2 nM (Fig 4C), which indicates an almost 3.4-fold enhancement of affinity compared with the inhibition constant of the single inhibitory unit of the APPI monomer ($K_i$ = 122.5±10.0 nM) (Fig 4B), whose $K_i$ value was consistent with previous studies [33, 36]. This enhancement in affinity may result from the mesotrypsin protein dimerizing upon binding the APPI dimer and the sequential binding of the two mesotrypsin active sites (in the mesotrypsin dimer) to the two APPI canonical binding loops (in the APPI dimer), as discussed in the next section.

### Kinetic model for APPI homodimer binding to mesotrypsin

To quantitatively understand the observed cooperativity in binding affinity (*i.e.*, avidity) for the APPI homodimer—mesotrypsin complex, we propose a simple kinetic model that goes beyond the standard Michaelis-Menten model for competitive enzyme inhibition (Fig 5). The proposed model assumes that the reaction mechanism consists of two consecutive steps. In the first step one mesotrypsin molecule reversibly binds a single APPI unit of the APPI homodimer. In the second step, the second mesotrypsin molecule reversibly binds the remaining

## Reversible Mesotrypsin Dimerization Induced by Homodimer APPI

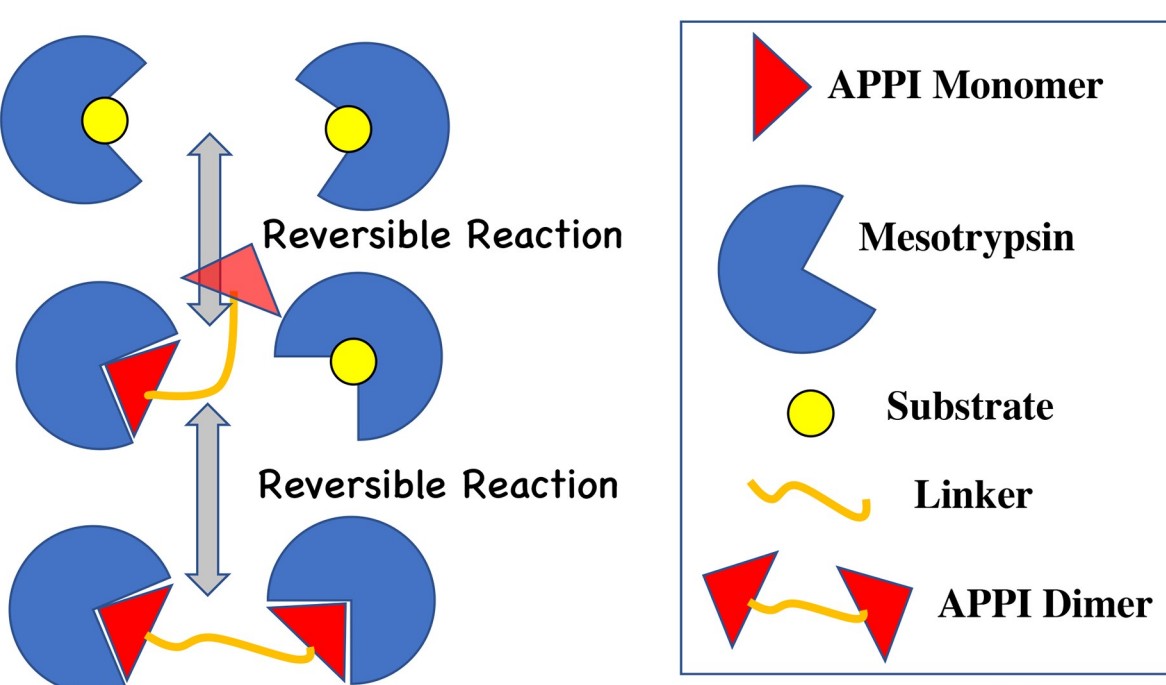

**Fig 5. Proposed reaction mechanism for the APPI homodimer induced mesotrypsin dimerization.** The mechanism allows two steps. In the first step, one APPI monomer unit reversibly binds one mesotrypsin molecule (and the second APPI unit remains unbound). In the second step, the remaining APPI unit reversibly binds the second mesotrypsin molecule.

APPI unit of the APPI homodimer. Such model is commonly termed in supramolecular chemistry as 'The 1:2 system' (see Ref. [45] for an extensive tutorial review).

The following parallel reactions take place in the system. The enzyme catalyzes its substrate via a standard Michaelis-Menten mechanism:

$$E + S \overset{k_a}{\Leftrightarrow} [ES] \overset{k_b}{\to} E + P \tag{1}$$

The reaction equation takes the form:

$$\frac{d[ES]}{dt} = k_a[E][S] - k_{a'}[ES] - k_b[ES] \tag{2}$$

where $[E]$ is the enzyme concentration, $[S]$ is the substrate concentration, and $[ES]$ is the concentration of the enzyme-substrate complex; $k_a$ and $k_{a'}$ are the association and dissociation rate constants of the enzyme-substrate binding reaction, respectively, and $k_b$ is the catalytic rate constant leading to product $[P]$. In equilibrium this reaction is characterized by the Michaelis-Menten constant $K_M$:

$$K_M = \frac{[E][S]}{[ES]} = \frac{k_{a'} + k_b}{k_a} \tag{3}$$

For APPI homodimers interacting with mesotrypsin enzyme molecules, the reaction mechanism allows two steps. In the first step, one APPI monomer unit reversibly binds one mesotrypsin enzyme molecule (and the second APPI unit remains unbound). In the second step, the remaining APPI unit reversibly binds the second mesotrypsin enzyme molecule (Fig 5). The first step of such mechanism is characterized by the reaction:

$$E + I \overset{\mathbf{K}_I}{\Leftrightarrow} [EI] \tag{4}$$

with the following reaction equation:

$$\frac{d[EI]}{dt} = k_1[E][I] - k_{-1}[EI] \tag{5}$$

where $k_1$ is the association rate constant of mesotrypsin molecule binding to one unit of APPI homodimer (when the second APPI unit remains unbound), and $k_{-1}$ is the corresponding dissociation rate constant; $[I]$ is the APPI concentration, $[EI]$ is the concentration of mesotrypsin enzyme molecules bound to a single APPI unit. In equilibrium this reaction is characterized by the equilibrium inhibition constant, $\mathbf{K}_I$:

$$\mathbf{K}_I = \frac{[E][I]}{[EI]} = \frac{k_{-1}}{k_1} \tag{6}$$

The second step in the APPI homodimer binding to mesotrypsin molecules (Fig 5) is described by the reaction:

$$[EI] + E \overset{\mathbf{K}_2}{\Leftrightarrow} [(2E)I] \tag{7}$$

with the following reaction equation:

$$\frac{d[(2E)I]}{dt} = k_2[EI][E] - k_{-2}[(2E)I] \tag{8}$$

where $k_2$ is the association rate constant for the binding of the remaining APPI unit of the

APPI homodimer to the second mesotrypsin enzyme molecule, and $k_{-2}$ is the corresponding dissociation rate constant; $[(2E)I]$ is the concentration of mesotrypsin enzyme molecule pairs in complex with APPI homodimers. In equilibrium this reaction is characterized by the equilibrium inhibition constant, $\mathbf{K}_2$:

$$\mathbf{K}_2 = \frac{[EI][E]}{[(2E)I]} = \frac{k_{-2}}{k_2} \tag{9}$$

Combining Eqs 6 and 9 we obtain:

$$\frac{[E]^2[I]}{[(2E)I]} = \mathbf{K}_2\mathbf{K}_\mathrm{I} \equiv \mathbf{K}_{2\mathrm{I}}^2 \tag{10}$$

where we defined a new equilibrium inhibition constant, $\mathbf{K}_{2\mathrm{I}}$.

We stress that the consecutive reactions, Eqs 4 and 7, are entirely equivalent to the following elementary reaction:

$$2E + I \overset{\mathbf{K}_{2\mathrm{I}}}{\Leftrightarrow} [(2E)I] \tag{11}$$

and the corresponding kinetic equation:

$$\frac{d[(2E)I]}{dt} = k_{2i}[E]^2[I] - k_{-2i}[(2E)I] \tag{12}$$

Therefore, in equilibrium this equation is characterized by the equilibrium inhibition constant, $\mathbf{K}_{2\mathrm{I}}$, which is identical to Eq 10:

$$\frac{[E]^2[I]}{[(2E)I]} = \frac{k_{-2i}}{k_{2i}} \equiv \mathbf{K}_{2\mathrm{I}}^2 \tag{13}$$

We stress the important fact that unlike the conventional Michaelis-Menten scheme for competitive enzyme inhibition, where the enzyme concentration enters via a linear term, $k_1[E][I]$, (Eq 5), here (Eq 12) the enzyme concentration enters via a non-linear term, $i.e.$, quadratic in the mesotrypsin concentration, $k_{2i}[E]^2[I]$.

Combining the mass conservation equation for mesotrypsin ($E_\mathrm{tot}$ is the total concentration of mesotrypsin in the system):

$$[E] + [ES] + [EI] + 2[(2E)I] = E_{tot} \tag{14}$$

with the equilibrium equations, Eqs 3, 6 and 10, we obtain the resulting equation for $[ES]$ (see S1 File for a step-by-step derivation):

$$[ES]^2 \frac{2[I]}{\mathbf{K}_{2\mathrm{I}}^2}\left(\frac{K_M}{[S]}\right)^2 + [ES]\left(1 + \frac{K_M}{[S]}\alpha\right) - E_{tot} = 0 \tag{15}$$

where $\alpha$ is the following:

$$\alpha = 1 + \frac{[I]}{\mathbf{K}_\mathrm{I}} \tag{16}$$

The solution of this equation:

$$[ES] = \frac{-\left(1 + \frac{K_M}{[S]}\alpha\right) + \sqrt{\left(1 + \frac{K_M}{[S]}\alpha\right)^2 + 8E_{tot}\left(\frac{K_M}{[S]}\right)^2 \frac{[I]}{K_{2I}^2}}}{4\left(\frac{K_M}{[S]}\right)^2 \frac{[I]}{K_{2I}^2}} \qquad (17)$$

and the resulting reaction velocity:

$$v = k_b[ES] \qquad (18)$$

where $k_b$ is the catalytic rate constant leading to product $[P]$.

Fitting Eq 18 to the experimental data, allows us to obtain the value of $K_{2I}$ (Fig 6). The fits to kinetic measurements performed at different APPI concentrations show an excellent agreement with the data and provide the average value for the equilibrium inhibition constant

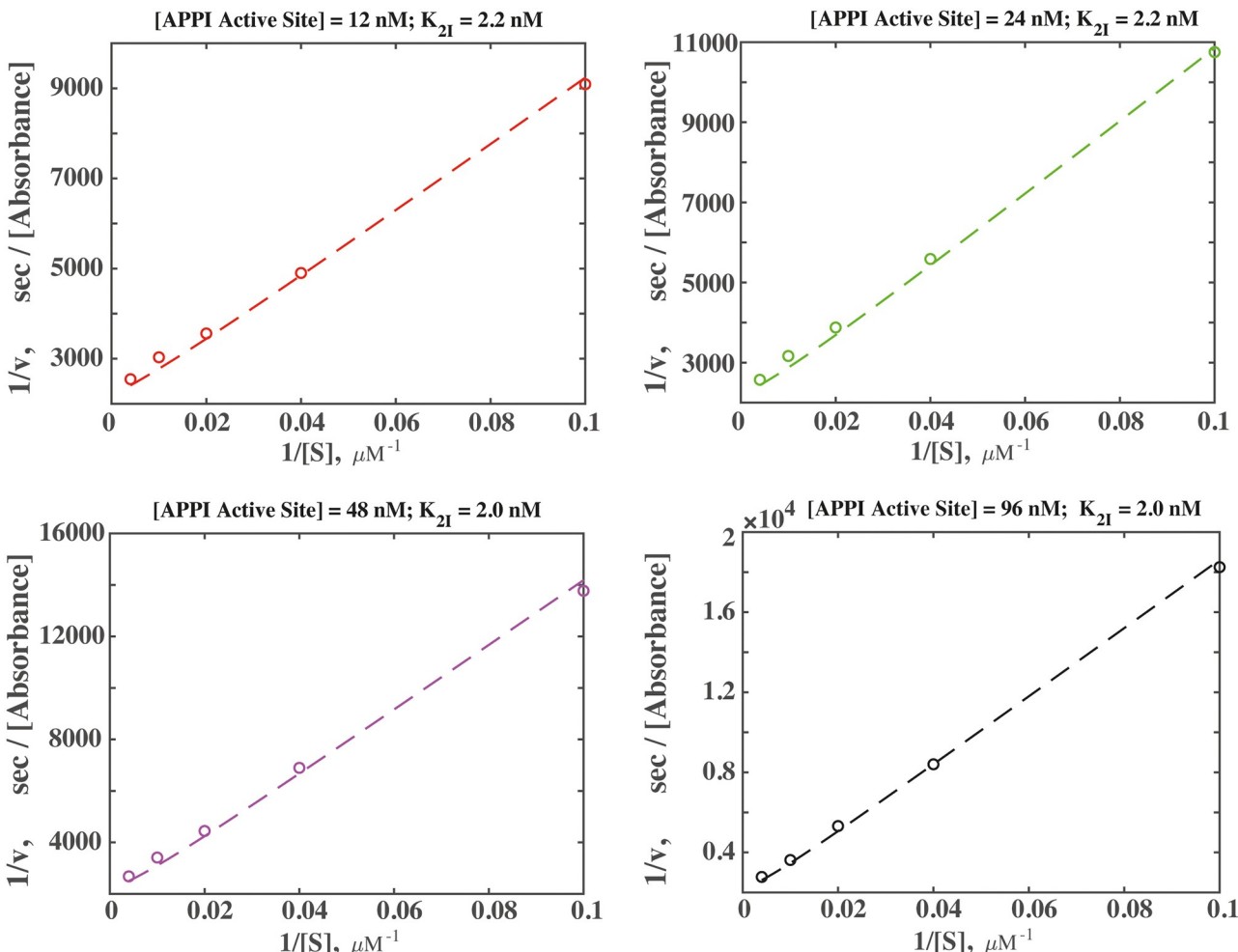

**Fig 6. Two-step model (Fig 5) for mesotrypsin inhibition by APPI homodimers provides an excellent fit to experimental data.** Fitting the reaction velocity (Eq 18) measured at different substrate concentration, allows us to extract the equilibrium inhibition constant $K_{2I}$. Each of the four plots shows the fit corresponding the measurements performed at a given APPI homodimer concentration $[I]$, where [APPI Active Site] = 2$[I]$, stands for the concentration of APPI monomer units. The added mesotrypsin concentration was maintained at $[E_{tot}]$ = 0.25 nM, and we adopted $K_I$ = 122.5 nM, as independently determined from the measurements performed with monomeric APPI (Fig 4B). The fitted values of $K_{2I}$ are presented above each plot. Raw data can be found in SI.

defined by Eq 10, $K_{2I} \approx 2.1$ nM. Intuitively, $K_{2I}$ represents a measure characterizing the equilibrium balance between two possible types of enzyme-APPI complexes: [$EI$] (*i.e.*, concentration of mesotrypsin enzyme molecules bound to a single APPI unit), and [$(2E)I$] (*i.e.*, the concentration of mesotrypsin enzyme molecule pairs in complex with APPI homodimers). We stress the fact that $K_{2I}$ cannot be directly compared to the corresponding equilibrium constant extracted above using a conventional Michaelis-Menten scheme ($K_i = 36.1$ nM, Fig 4C), since $K_{2I}$ is obtained using a different kinetic mechanism. Intuitively, the observed cooperativity in binding affinity (*i.e.*, avidity) of the APPI homodimer, as compared to the APPI monomer, stems from the fact that the APPI homodimer—mesotrypsin association rate, $k_{2i}[E]^2[I]$, scales as a second power of the enzyme concentration, leading to a more efficient inhibition. This is unlike the APPI monomer case, where the APPI—mesotrypsin association rate, $k_1[E]$ [$I$], scales linearly with enzyme concentration; here $k_1$ is the association rate constant of the mesotrypsin—monomer APPI binding reaction.

Based on the determined value of $K_{2I} = 2.1$ nM, using Eq 10, it is now possible to extract the equilibrium inhibition constant for the second reaction step of mesotrypsin binding to the APPI homodimer, $K_2$: $K_2 = K_{2I}^2/K_I$, where we use $K_I = 122.5$ nM is the measured equilibrium inhibition constant for the APPI monomer (Fig 4B). Here we assume that such value of $K_I$ measured for the case of monomeric APPI, also represents the equilibrium inhibition constant for the first reaction step (Eq 4) in the case of homodimer APPI. This gives us, $K_2 = 0.036$ nM. Such strikingly low value of $K_2$ as compared to $K_I$ (where $K_2$ is nearly four orders of magnitude lower than $K_I$) indicates a high degree of cooperativity (avidity) in the second step of the reaction mechanism, where the second mesotrypsin molecule binds to the remaining unit of APPI homodimer, as compared to the first step, where just a single mesotrypsin molecule binds to a single unit of APPI homodimer (Fig 5). Such low value of $K_2$ indicates that the equilibrium balance in our system is shifted towards the second reaction step.

The increase in affinity measured here for a dimeric compared with a monomeric inhibitor–enzyme complex is much smaller than that known for polyvalent compared with monovalent antibody–antigen complexes. In our study, avidity occurs in the context of the interaction between a bivalent inhibitor and a monomeric enzyme with a single active site, where the enzyme is in solution and unclustered prior to binding to the bivalent inhibitor.

By contrast, in antibody–antigen complexes, the antigen (or receptor) molecules are anchored to a surface (e.g., to a cell membrane) and thus are clustered in close proximity to each other before they bind the multivalent antibody ligand. Such clustering significantly enhances interaction strength, such that affinity in polyvalent antibody–antigen complexes is 2–3 orders of magnitude stronger than in their monovalent analogues [46, 47]. When monomeric target molecules with a single binding site are in solution, the molecules are separated from each other and have greater mobility than clustered (or surface anchored) target molecules and binding sites. This may make the binding of the monomeric targets to a dimeric ligand more difficult if they follow a 'lock and key' mode of molecular recognition. In contrast, during interactions with antibodies, the organization of surface-immobilized targets exhibits a high surface epitope density that allows them to rebind the multivalent ligand much faster and in the correct conformation for optimal binding. This point was recently demonstrated by Hadzhieva *et al.* [40] in a study that elucidated the correlation between the binding affinities of several IgG antibodies and the density of an antigen attached to a surface. The authors chose several antibodies that bound the HIV envelope glycoprotein 120 (namely gp120) with a wide range of affinities. Using a surface plasmon resonance (SPR) spectroscopy method to evaluate affinity, they controlled the density of antigen immobilized to the surface of a sensor chip and performed avidity measurements. As antigen density increased, enhanced avidity was

observed for all the antibodies, as expressed by a systematic decrease in their equilibrium dissociation constants driven largely by reductions in their dissociation rate constants. For example, for the HJ16 antibody, a 60-fold increase in gp120 density led to a 150-fold decrease in the dissociation rate constant that was expressed as a 168-fold increase in binding affinity. One of the proposed explanations for this result was that the high density of gp120 allowed the formation of an IgG hexametric structure, which resulted in enhanced avidity [40]. Taken together, it is plausible that a greater avidity effect is observed in systems that confer both high target density and restricted target movement (as in surface-immobilized antigens or membrane-anchored receptors). This assumption is supported by the fact that proper orientation of free target molecules is a prerequisite for optimal ligand binding. The formation of a soluble complex, such as in interactions between an inhibitor and its target enzyme, requires the specific orientation of the interface residues of both interacting proteins. Obtaining the optimal orientation for binding is much more difficult for soluble complexes because of the dynamic and constant movement of the target molecules in solution, whereas immobilized targets with restricted movements may confer faster association rates and higher affinity in equilibrium.

In other words, the binding epitopes of a surface-anchored protein target may be subject to conformational constraints that may result in lower entropic penalties upon binding to a ligand, which enhances binding affinity [5, 48]. For monomeric protein targets in solution, the structure of the target is flexible, such that there is a large difference in the structural rigidity of the protein target between its unbound state and its state when bound to its polyvalent ligand. In contrast, the structure of the target is more rigid when immobilized to a surface, such that there is little difference in rigidity between its unbound and bound states. This conformational constraint may also be achieved by target dimerization, as observed for mesotrypsin in our system.

The positive cooperativity we observed between APPI units binding to mesotrypsin stands in contrast to the results obtained by Farooq's group, who characterized ligand binding to the two WW tandem domains of the YAP2 protein, a transcriptional factor that regulates genes related to cell fate [12]. They isolated the WW domains and determined the binding affinity of each WW domain toward each ligand from equilibrium dissociation constants using isothermal titration calorimetry measurements. Next, they compared the constants obtained for a specific ligand with the binding affinity of the same ligand toward the native structure (*i.e.*, to YAP2 containing two WW domains). They found negative cooperativity during ligand binding to tandem WW domains. Specifically, each ligand exhibited higher affinity for the isolated WW domain in comparison with the tandem, fused WW domains of the native YAP2 protein. The researchers explained this negative cooperativity by positing that inter-domain interactions arising from the proximity of the two WW domains to each other impede or sterically hinder ligand binding to each domain.

Bikunin (like APPI) is a member of the Kunitz family, however (unlike APPI) it has two subunits that are linked via a single peptide bond. Similarly to the WW domain in YAP2, bikunin cannot bind two serine protease simultaneously. Ollis's group postulated that the native structure of bikunin can control the proteolytic activity of enzymes because each domain binds a different protease and so acts as a useful regulatory mechanism, with one Kunitz domain diminishing or abolishing binding at the second domain [37].

These previous studies and our own current study include examples of soluble systems that have two domains within a single protein ligand, with each domain having a single binding site capable of interacting with monomeric protein targets in solution. However, the major difference between our study and the previous studies of bikunin and the WW domains in YAP2 is that their tandem domains, when embedded in the entire protein, do not exhibit enhanced binding affinity, with the lack of enhancement attributed to steric interference. On the other

hand, in our system the affinity of the target molecule mesotrypsin to a single binding loop in dimeric APPI was enhanced relative to the binding of mesotrypsin to the same binding loop in monomeric APPI. This was probably because there was negligible steric interference between the single APPI units (and specifically the canonical inhibitory loops) within the dimeric APPI structure. In the absence of steric hindrance, our dimeric APPI could simultaneously bind, via its canonical binding loops, two mesotrypsin units as shown in our APPI dimer titration analysis. In addition, our dimeric APPI protein is a homodimer, whereas the WW domains of the YAP-2 protein and the two Kunitz domains of bikunin differ in their amino acid content (*i.e.*, they are heterodimers) which may also influence their target binding kinetics.

## Conclusions

In summary, we have demonstrated the feasibility of enhancing binding affinity through an avidity effect using a complex between mesotrypsin and bivalent APPI, in which the mesotrypsin target protein is monomeric in solution (prior to binding) and has a single binding site. Such enhanced binding affinity (*i.e.*, avidity) stems from statistically enhanced attraction between pairs of mesotrypsin molecules reversibly binding to the homodimer APPI. Isolation of the fully active recombinant dimeric APPI exhibiting correct folding and possessing two fully functional inhibitory loops (one on each monomer) allowed us to achieve accurate results, unaffected by unfolded or partially unfolded protein domains. Within the framework of conventional Michaelis-Menten model for competitive inhibition, the binding affinity, which was measured for a single inhibitory loop, was 3.4-fold stronger for dimeric APPI compared with monomeric APPI. Strikingly, revising such conventional Michaelis-Menten kinetic model, and introducing a non-linear effect induced by the presence of bivalent APPI homodimer, we obtain a nearly four orders of magnitude stronger binding affinity in the second step of the reaction mechanism (two mesotrypsin molecules binding to the APPI homodimer), as compared to the first step (a single mesotrypsin molecule binding to a single APPI unit of the APPI homodimer), Fig 5. This model provides a quantitative explanation for the observed enhanced tendency for mesotrypsin dimerization leading to cooperativity (*i.e.*, avidity) in the binding affinity that cannot be explained by the conventional Michaelis-Menten scheme. Moreover, we validated the ability of dimeric APPI to bind two mesotrypsin units by using binding titration analysis, indicating that the enzyme recognition site on each of the two inhibitory units possesses a canonical loop that is available for simultaneous interaction with trypsins.

## Materials and methods

### Reagents

Synthetic oligonucleotides were obtained from Integrated DNA Technologies (Coralville, IA, USA). Restriction enzymes, T4 ligase, and Q5 polymerase were purchased from New England Biolabs (Ipswich, MA, USA), and nucleoside triphosphates (dNTPs) from Jena Bioscience (Jena, Germany). The methylotrophic yeast *P. pastoris* strain GS115 and the Pichia expression vector (pPIC9K) were obtained from Invitrogen (Carlsbad, CA, USA). Bovine trypsin, disuccinimidyl suberate (DSS), and the chromogenic substrates benzyloxycarbonyl-Gly-Pro-Arg-p-nitroanilide (Z-GPR-pNA), 4-nitrophenyl 4-guanidinobenzoate (pNPGB), and benzoyl-L-arginine-p-nitroanilide (L-BAPA) were purchased from Sigma-Aldrich (St. Louis, MO, USA). Rabbit anti-trypsin antibody (Ab-200997) and mouse anti-his tag (Ab-49936) were obtained from Abcam (Cambridge, UK). Affi-gel 10 resin was obtained from Bio-Rad laboratories (Hercules, CA, USA).

## Generation of APPI gene constructs in expression vector pPic9k

APPI monomer (PDB:3L33, residues 4–55) was cloned into the plasmid pPic9k as previously described [36] and the gene construct encoding dimeric APPI was generated by a gene assembly method. Briefly, pPic9k plasmid containing the full-length APPI gene was amplified by polymerase chain reaction (PCR) using Q5 DNA polymerase with primers containing appropriate restriction sites for pPic9k plasmid and the addition of GGGGS×3 linker at the C-terminal of APPI unit. The following primers were used to amplify the monomeric APPI: APPI FW: (5′-TGCTACGTATTAATTAACGAAGTTTGTTCTGAACAAGCTG-3′) and APPI RC: (5′-GCAATGGAATTCGGATCCCCCTCCTCCGGATCCTCCCCCTCCGGAACCTCCCCCTCCAATAGCAGAACCACAAACAGC-3′).

Both the amplified PCR fragments and pPick9k plasmid were digested with the SnaBI and EcoRI restriction enzymes to generate an APPI dimer gene construct (S1A Fig in S1 File). The digested fragments were ligated using T4 ligase, transformed into *E. coli*, and plated on plates with lysogeny broth containing ampicillin (LB-amp). Finally, the dimeric APPI gene construct was sequenced at the DNA Microarray and Sequencing Unit of the National Institute for Biotechnology in the Negev at Ben Gurion University (DMSU, NIBN, BGU). Expression vectors were linearized by SacI digestion and used to transform *P. pastoris* strain GS115 by electroporation. This resulted in the insertion of the construct at the first alcohol oxidase (AOX1) locus of *P. pastoris*, thereby generating a His+ Mut+ phenotype. Transformants were selected for the His+ phenotype on 2% agar containing regeneration dextrose biotin (RDB; 18.6% sorbitol, 2% dextrose, 1.34% yeast nitrogen base, $4\times10^{-5}$% biotin, and 0.005% each of L-glutamic acid, L-methionine, L-lysine, L-leucine, and L-isoleucine) and allowed to grow for 3 days at 30˚C. Cells were harvested from the plates and subjected to further selection for high copy number on the basis of their growth on 2% agar containing 1% yeast extract, 2% peptone, 2% dextrose medium, and the antibiotic G418 (Geneticin, 4 mg/mL, Invitrogen). To verify direct insertion of the construct at the AOX1 locus of *P. pastoris*, the genomic DNA of the highest APPI-expressing colony from each APPI variant was extracted and amplified by PCR with an AOX1 upstream primer, 5′- GACTGGTTCCAATTGACAAGC-3′, and an AOX1 downstream primer, 5′- GCAAATGGCATTCTGACATCC-3′ (S1B Fig in S1 File). The PCR products were separated on 1% agarose gel, purified, and the correct sequence of the APPI dimer was confirmed by DNA sequencing analysis (DMSU, NIBN, BGU) (S1C Fig in S1 File).

## Large-scale purification of APPI variants

*P. pastoris* cultures expressing either monomeric or dimeric APPI were grown in 50 mL of BMGY medium (1% yeast extract, 2% peptone, 0.23% $KH_2PO_4$, 1.18% $K_2HPO_4$, 1.34% yeast nitrogen base, $4\times10^{-5}$% biotin, and 1% glycerol) overnight and then in 0.5 L of BMMY medium (similar to BMGY, but with 0.5% methanol instead of 1% glycerol) for 3 days, with 2% methanol being added every 24 h to maintain induction. Following 4 days of induction, the culture was centrifuged at 3800×g for 10 min and the supernatant containing the secreted recombinant inhibitors was prepared for purification by nickel-immobilized metal affinity chromatography (IMAC). The supernatant was adjusted to 10 mM imidazole and 0.5 M NaCl at pH 8.0 and left to stand for 1 h at 4˚C. Thereafter, filtration was performed to remove any additional precipitation using a 0.22 μm Steritop bottle-top filter (Millipore, MA, USA). The filtered supernatant was loaded on a HisTrap 5 mL column (GE Healthcare, Piscataway, NJ) at a flow rate of 0.5 mL/min for 24 h, washed with a washing buffer (20 mM sodium phosphate 0.5 M NaCl, and 10 mM imidazole (pH 8.0)), and eluted with an elution buffer (20 mM sodium phosphate, 0.5 M NaCl, and 0.5 M imidazole) in an ÄKTA-Pure instrument (GE Healthcare, Piscataway, NJ). The buffer of the eluted proteins was replaced with a mesotrypsin

buffer (100 mM tris(hydroxymethyl)aminomethane (Tris), 1 mM CaCl$_2$ (pH 8.0)) using a 3.5 kDa cutoff dialysis kit (Gene Bio Application, Israel). For all recombinant proteins (both monomeric and dimeric APPI), gel filtration chromatography was performed using a Super-dex 75 16/600 column (GE Healthcare, Piscataway, NJ) equilibrated with a mesotrypsin buffer (100 mM Tris, 1mM CaCl$_2$ (pH 8.0)) at a flow rate of 1 mL/min on an ÄKTA-Pure instrument (GE Healthcare, Piscataway, NJ). The purified proteins were analyzed by sodium dodecyl sul-phate–polyacrylamide gel electrophoresis (SDS-PAGE) on a 15% polyacrylamide gel under reducing conditions followed by staining with InstantBlue Coomassie Protein Stain (Expe-deon, Cambridge, UK). Finally, the correct mass of the proteins was validated using a Matrix-Assisted Laser Desorption/Ionization-Time of Flight (MALDI-TOF) REFLEX-IV (Bruker) mass spectrometer (Ilse Katz Institute for Nanoscale Science & Technology, BGU; S2 Fig in S1 File).

## Preparation of the trypsin column

Affi-gel 10 beads (8 mL) were mixed with isopropanol, transferred into a 15 mL falcon tube, and centrifuged at 1000 rpm for 10 min. Then, the isopropanol was discarded, and the beads were washed with cold double-distilled water (DDW; 10 mL) and centrifuged at 2000 rpm for 10 min. This procedure was repeated three times to remove the isopropanol residues for effi-cient coupling of bovine trypsin to the beads. Thereafter, bovine trypsin powder (29.5 mg) was dissolved in 100 mM HEPES pH 7.5 (3 mL) and added to the falcon tube that contained the resin. The mixture was left rotating overnight at 4˚C. The conjugated beads were then packed into a 9 mL glass column (Bio-Rad laboratories, Hercules, CA, USA). The dimeric APPI (15 mL) was loaded onto the column and then eluted with a gradient of 0–100% using 100 mM HCl.

## Enzymes and substrates

Recombinant human mesotrypsinogen was expressed in *E. coli*, extracted from inclusion bod-ies, refolded, purified, and activated with bovine enteropeptidase, as described previously [49, 50]. The concentrations of mesotrypsin and bovine trypsin were determined by active site titration using p-nitrophenyl 4-guanidinobenzoate hydrochloride (pNPGB) substrate (Sigma-Aldrich, St. Louis, MO, USA), which served as an irreversible inhibitor [51]. For determination of substrate concentration, an aliquot of the chromogenic substrate N-α-benzoyloxycarbonyl-glycylprolylarginine p-nitroanilide (Z-GPR-pNA) dissolved in dimethyl sulfoxide (DMSO) was incubated with an excess of bovine trypsin powder to obtain full substrate cleavage. The substrate concentration was determined from change in absorbance at 410 nm as a result of the release of p-nitroaniline ($\varepsilon_{410}$ = 8480 M$^{-1}$ cm$^{-1}$).

## Determination of the inhibitory unit concentration by trypsin titration

In the canonical APPI monomer, the enzyme recognition site is a single loop, whereas in a fully functional APPI homodimer, enzyme binding should involve two loops. Since APPI inhibits the activity of serine protease mesotrypsin, each APPI loop or binding site may be referred to as an inhibitory unit. To accurately measure the concentration of inhibitory units available to interact with the protease in the APPI monomer and dimer, titration was per-formed with pre-titrated bovine trypsin and Nα-benzoyl-L-arginine 4-nitroanilide hydrochlo-ride (L-BAPA; Sigma-Aldrich, St. Louis, MO, USA), as described previously [49]. Briefly, an assay cocktail (312 μl; 100 Mm Tris, 5 mM CaCl$_2$ (pH = 8.0)) having a final concentration of 100 μM L-BAPA was added into a 96-well microplate (Greiner, Kremsmünster, Austria). Bovine trypsin enzyme (9 μL, 30 μM) was mixed with each of six dilutions of APPI monomer

or dimer to produce six enzyme–inhibitor mixtures (final volume 45 μL). An aliquot (8 μL) of each enzyme–inhibitor mixture was transferred into the wells of a 96-well microplate to initiate the reaction. Reactions were monitored spectroscopically at 410 nm for 5 min at 37˚C. The reaction velocity (being the change in absorbance with time during substrate cleavage) was calculated for each inhibitor concentration, and the values were plotted versus the volume of inhibitor (0–30 μL APPI) in the enzyme–inhibitor mixture. From these data, the X-intercept was determined, and the inhibitory unit concentrations were calculated using Eq 19. Reported inhibitory unit concentrations are the average values obtained from two independent experiments, reported as mean±SD.

$$[IU] = DF_i V_e \frac{[E]}{X_{int}} \tag{19}$$

Where [IU] is the inhibitory unit concentration (μM) of the APPI monomer or dimer, $DF_i$ is the inhibitor dilution factor (11.5 and 1.7 dilution from the monomeric and dimeric APPI stock solution, respectively), $V_e$ is enzyme volume (*i.e.*, 9 μL bovine trypsin), [E] is enzyme concentration (*i.e.*, 30 μM), and $X_{int}$ is the value of the x-intercept on a plot of reaction velocity (AU/s) versus inhibitor (APPI) volume (μL).

## Inhibition studies

The inhibition constants ($K_i$) of monomeric and dimeric APPI in complex with mesotrypsin were determined according to the previously described methodology, with minor changes [49]. Briefly, stock solutions of enzyme, substrate, and APPI monomer (or dimer) were prepared at 40× the desired final concentrations. Assays were performed at 37˚C in the presence of different concentrations of Z-GPR-pNA substrate (5–250 μM) and inhibitor (APPI monomer, 0–400 nM; APPI dimer inhibitor units, 0–96 nM (APPI dimer molecule = 0–48 nM to achieve 2× this concentration of inhibitor units) in a Synergy 2 Multi-Detection Microplate Reader (BioTek, VT, USA). Mesotrypsin buffer (296 μL), Z-GPR-pNA (8 μL), and diluted APPI monomer/dimer (8 μL) were mixed and equilibrated in a 96-well microplate (Greiner, Kremsmünster, Austria) prior to the addition of mesotrypsin (8 μL from 10 nM stock). Reactions were followed spectroscopically for 5 min, and initial rates were determined from the increase in absorbance caused by the release of p-nitroaniline. Data were globally fitted by multiple regression to Eq 20, the classic competitive inhibition equation, using Prism (Graph-Pad Software, San Diego, CA).

$$u = \frac{k_{cat}[E]_0[S]}{K_m(1 + [I]/K_i) + [S]} \tag{20}$$

Where $u$ is the velocity of product formation at the start of the reaction; $K_m$ (the Michaelis-Menten constant) and $k_{cat}$ are the kinetic parameters for substrate hydrolysis; $[E]_0$ and $[I]$ are the total concentrations of enzyme and inhibitor, respectively, and $[S]$ is the initial substrate concentration. The reactions were performed with excess APPI monomer and dimer inhibitor unit concentrations ($\geq 20$ times the mesotrypsin concentration), and therefore any reduction of inhibitor concentration upon binding was negligible. Reported inhibition constants are average values obtained from three independent experiments, and are expressed as mean±SD.

## Supporting information

**S1 Raw images. 15% SDS-PAGE analysis of the purified monomeric APPI fraction (A) and comparison of purified monomeric APPI with dimeric APPI before and after purification**

**(B).** This figure shows the original gels presented in Figs 1E and 2C, respectively.
(DOCX)

**S1 Data.**
(XLSX)

**S1 File.**
(DOCX)

## Acknowledgments

The authors thank Dr. Itay Cohen for technical assistance and advice.

## Author Contributions

**Conceptualization:** Shiran Lacham-Hartman, David B. Lukatsky, Niv Papo.

**Data curation:** Shiran Lacham-Hartman, David B. Lukatsky.

**Formal analysis:** Shiran Lacham-Hartman, Yulia Shmidov, Ronit Bitton, David B. Lukatsky, Niv Papo.

**Funding acquisition:** Evette S. Radisky, David B. Lukatsky, Niv Papo.

**Investigation:** Shiran Lacham-Hartman, Yulia Shmidov, David B. Lukatsky, Niv Papo.

**Methodology:** Shiran Lacham-Hartman, Evette S. Radisky, Ronit Bitton, David B. Lukatsky.

**Project administration:** Niv Papo.

**Resources:** Evette S. Radisky, David B. Lukatsky, Niv Papo.

**Supervision:** Niv Papo.

**Validation:** Shiran Lacham-Hartman, David B. Lukatsky, Niv Papo.

**Visualization:** Niv Papo.

**Writing – original draft:** Shiran Lacham-Hartman, Niv Papo.

**Writing – review & editing:** Yulia Shmidov, Evette S. Radisky, Ronit Bitton, David B. Lukatsky.

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
