## [Decision Letter · Decision Letter 0]

12 Apr 2021

PONE-D-21-08844

Avidity observed between a bivalent inhibitor and an enzyme monomer with a single active site

PLOS ONE

Dear Dr. Papo,

Thank you for submitting your manuscript to PLOS ONE. After careful consideration, we feel that it has merit but does not fully meet PLOS ONE’s publication criteria as it currently stands. Therefore, we invite you to submit a revised version of the manuscript that addresses the points raised during the review process.

In your revised manuscript, please address as fully as possible the comments of the two reviewers - in particular the detailed and constructive comments of Reviewer 2.

We look forward to receiving your revised manuscript.

Kind regards,

Israel Silman

Academic Editor

PLOS ONE

Journal Requirements:

Reviewers' comments:

Reviewer's Responses to Questions

**Comments to the Author**

1. Is the manuscript technically sound, and do the data support the conclusions?

Reviewer #1: Yes

Reviewer #2: No

2. Has the statistical analysis been performed appropriately and rigorously? 

Reviewer #1: N/A

Reviewer #2: No

3. Have the authors made all data underlying the findings in their manuscript fully available?

Reviewer #1: Yes

Reviewer #2: No

4. Is the manuscript presented in an intelligible fashion and written in standard English?

Reviewer #1: Yes

Reviewer #2: Yes

5. Review Comments to the Author

Reviewer #1: Lacham-Hartman et al. show avidity benefits for a bivalent ligand with a soluble monomeric target, as opposed to the more widely studied systems involving multiple copies of both receptors and ligands. The work is novel, interesting and certainly adds to the knowledge in the field of multivalency.

The manuscript is concisely written and is easy to understand. The authors do a good job at providing enough background of prior work to the reader. The manuscript clearly articulates what the current work adds to the field. Further, the authors also cite and explain other contrasting theories in light of their findings.

Please consider the following minor revisions before publication-

1) In the proposed model in Fig 5, have the authors considered an alternate scenario (or rather a parallelly occuring one) where an APPI dimeric unit inhibits a single mesotrypsin unit – binding to the same mesotrypsin alternating between the two units of the engaged APPI homodimer. This scenario would still offer the proposed benefits of avidity. How does this scenario change the analysis proposed by the authors?

2) Could the authors clarify how they determined 3 repeats of GGGGS to be appropriate for ‘the simultaneous binding to two mesotrypsin units’? (To prevent any antagonistic effects due to steric hindrance, one could rationally determine the length of the spacers: looking at the APPI protein structure, the location of the inhibitory loop, and the terminus chosen to attach the second unit of APPI in tandem. The following review is a good source for fusion protein linkers and estimated lengths for future reference: doi.org/10.1016/j.addr.2012.09.039)

3) In Fig 1D left peak has a little bump which might explain the presence of higher order aggregates.

4) Fig 5 scheme has a typo: “Reversable” -> “Reversible”. Fig 5 caption has a typo: “Ka is the associate rate….” -> “Ka is the association rate…”.

Reviewer #2: The Papo group examines here if a bivalent inhibitor displays avidity towards its monomeric enzyme target. Understanding avidity, or as it is known in chemistry, multivalency, is one of the most important and interesting challenges yet to be fully solved when it comes to understanding intermolecular interactions. It is, as the authors point out, of particular importance when it comes to various biological process as well as in other fields including medical chemistry and nanoscience.

The key claim of this paper seems to be that the authors have uncovered a new insight into avidity through a very simple system; a bivalent inhibitor interacting with its monomer enzyme target.

I am really not certain if this is correct and whether the paper should be published will in my opinion come down to if the authors can make a more convincing case for this claim and make the connections of this work clearer with the current understanding of the field of avidity/multivalency, particularly from within the field of supramolecular chemistry. I will expand on these concerns in more details below:

(see attachment)

6. PLOS authors have the option to publish the peer review history of their article (what does this mean?). If published, this will include your full peer review and any attached files.

Reviewer #1: No

Reviewer #2: No

---

## [Author Response · Author response to Decision Letter 0]

14 Sep 2021

All page numbers written in the letter refer to the revised (with the track changes) manuscript.

Journal Requirements:

Q1. Please ensure that your manuscript meets PLOS ONE's style requirements, including those for file naming. The PLOS ONE style templates can be found at

A1. Done as requested.

Q2. PLOS ONE now requires that authors provide the original uncropped and unadjusted images underlying all blot or gel results reported in a submission’s figures or Supporting Information files. This policy and the journal’s other requirements for blot/gel reporting and figure preparation are described in detail at https://journals.plos.org/plosone/s/figures#loc-blot-and-gel-reporting-requirements and https://journals.plos.org/plosone/s/figures#loc-preparing-figures-from-image-files. When you submit your revised manuscript, please ensure that your figures adhere fully to these guidelines and provide the original underlying images for all blot or gel data reported in your submission. See the following link for instructions on providing the original image data: https://journals.plos.org/plosone/s/figures#loc-original-images-for-blots-and-gels.

A2. Done as requested. The original gel images of Fig. 1E and 2C are now in Supporting Information section (in Fig. S2). Please note that Figs 1F and 2B were already uncropped in the original figures. 

Q3. We note that you have included the phrase “data not shown” in your manuscript. Unfortunately, this does not meet our data sharing requirements. PLOS does not permit references to inaccessible data. We require that authors provide all relevant data within the paper, Supporting Information files, or in an acceptable, public repository. Please add a citation to support this phrase or upload the data that corresponds with these findings to a stable repository (such as Figshare or Dryad) and provide and URLs, DOIs, or accession numbers that may be used to access these data. Or, if the data are not a core part of the research being presented in your study, we ask that you remove the phrase that refers to these data.

A3. Done as requested. Please see revised parts on pages 7 and 9. 

 

Reviewer #1:

Lacham-Hartman et al. show avidity benefits for a bivalent ligand with a soluble monomeric target, as opposed to the more widely studied systems involving multiple copies of both receptors and ligands. The work is novel, interesting and certainly adds to the knowledge in the field of multivalency.

The manuscript is concisely written and is easy to understand. The authors do a good job at providing enough background of prior work to the reader. The manuscript clearly articulates what the current work adds to the field. Further, the authors also cite and explain other contrasting theories in light of their findings.

Please consider the following minor revisions before publication-

Q1. In the proposed model in Fig 5, have the authors considered an alternate scenario (or rather a parallelly occurring one) where an APPI dimeric unit inhibits a single mesotrypsin unit – binding to the same mesotrypsin alternating between the two units of the engaged APPI homodimer. This scenario would still offer the proposed benefits of avidity. How does this scenario change the analysis proposed by the authors?

A1. We thank Reviewer 1 for her/his important suggestion regarding the model. A similar suggestion was also pointed out by Reviewer 2. As a response to this comment, we revised the model, incorporating a two-step reaction mechanism in section ‘Kinetic model for APPI homodimer binding to mesotrypsin’ (see p. 15-20, the changes tracked, and Conclusions, p. 22). In the framework of the revised model (see new Fig. 5 of the revised manuscript), the reaction mechanism allows two steps. In the first step, one APPI monomer unit reversibly binds one mesotrypsin molecule (and the second APPI unit remains unbound). In the second step, the remaining APPI unit reversibly binds the second mesotrypsin molecule. Introducing two steps in the reaction mechanism, addresses the issue pointed out by Reviewer 1. The revised model leads to qualitatively similar results compared to the original model (new Fig. 6). Quantitatively, introducing the two-step model leads to ~10% increase in the resulting equilibrium constant, ��I (i.e., slightly decreased cooperativity (avidity)), as compared to the original one-step model (Fig. 6). The reason for the observed ~10% decrease in cooperativity stems from the fact that the second step of the two-step reaction mechanism, bringing two mesotrypsin molecules together through binding to APPI homodimer, constitutes the main source for cooperativity (see also the first paragraph on p. 18 of the revised manuscript).

Q2. Could the authors clarify how they determined 3 repeats of GGGGS to be appropriate for ‘the simultaneous binding to two mesotrypsin units’? (To prevent any antagonistic effects due to steric hindrance, one could rationally determine the length of the spacers: looking at the APPI protein structure, the location of the inhibitory loop, and the terminus chosen to attach the second unit of APPI in tandem. The following review is a good source for fusion protein linkers and estimated lengths for future reference: doi.org/10.1016/j.addr.2012.09.039)

A2. We thank the reviewer for the comment. The mesotrypsin-APPI heterodimer is shaped like a mushroom, where mesotrypsin is the mushroom cap (~40 Å in diameter) and APPI is the mushroom stem (~20 Å wide and ~30 Å high). Both the N- and C-termini of APPI are located opposite to its inhibitory loop, at the base of the molecule. By connecting 2 APPI molecules in tandem, we connect two mushroom-shaped complexes at the base of their stems, such that the two APPI subunits themselves can serve as rigid spacer modules of ~25 Å each, connected by the flexible linker hinge. Thus, even a short linker would be expected to allow >50 Å between the two mesotrypsin molecules resulting in little risk of steric hindrance between mesotrypsin subunits. Our larger concern was achieving efficient independent folding and correct disulfide bond configuration within each of the two APPI domains, and toward this end we selected a linker of generous length (i.e., 3 repeats of GGGGS, ~48 Å in length). Additionally, we isolated correctly folded APPI-dimer from misfolded species using affinity chromatography. We have now attempted to better clarify in the manuscript the reasoning behind the linker design (page 6).

Q3. In Fig 1D left peak has a little bump which might explain the presence of higher order aggregates.

A3. We agree with the reviewer and as can be seen in figure 2C (and also in the original gel, new Fig. S2B) upon using a trypsin column, these aggregates were removed.

Q4. Fig 5 scheme has a typo: “Reversable” -> “Reversible”. Fig 5 caption has a typo: “Ka is the associate rate….” -> “Ka is the association rate…”.

A4. Revised as requested. 

 

Reviewer #2:

The Papo group examines here if a bivalent inhibitor displays avidity towards its monomeric enzyme

target. Understanding avidity, or as it is known in chemistry, multivalency, is one of the most

important and interesting challenges yet to be fully solved when it comes to understanding

intermolecular interactions. It is, as the authors point out, of particular importance when it comes to

various biological process as well as in other fields including medical chemistry and nanoscience.

The key claim of this paper seems to be that the authors have uncovered a new insight into avidity

through a very simple system; a bivalent inhibitor interacting with its monomer enzyme target.

I am really not certain if this is correct and whether the paper should be published will in my opinion

come down to if the authors can make a more convincing case for this claim and make the

connections of this work clearer with the current understanding of the field of avidity/multivalency,

particularly from within the field of supramolecular chemistry. I will expand on these concerns in

more details below:

Q1. Part of the problem with this work is that researchers in biology/biophysics and those in

(supramolecular) chemistry use a completely different language about what are essentially the same

things – the (almost) synonyms of avidity and multivalency being a prime example. Usually what

happens then is that researchers will go to extreme lengths to make it appear that there are some

real differences between these terms and again, in the literature there are some very interesting

examples of authors trying to make it so that avidity and multivalency are different things when

there is little substance behind these terms. With this in mind, I would like to start to suggest to the

authors that they have good look at the chemistry literature on multivalency, e.g. from authors such

as Gianfranco Ercolani, George Whitesides, Pall Thordarson and Harry Anderson to mentioned a just

few (see c.f. Angew Chem 1998, 37, 2754, J. Phys. Chem. B, 2007, 111, 12195, Angew. Chem. 2009,

48, 7488, Angew. Chem., 2011, 50, 1762, Chem. Soc. Rev. 2017, 46, 2622, Bioconjugate Chem. 2019,

30, 503). This might obviously have some impact on their introduction to the topic.

A1. We thank the reviewer for this suggestion. Please see our additions (including references) to the Introduction section that better contextualize our current results against the background of the prior chemistry literature and its definitions of avidity and multivalency. In the added text, we identify the differing definitions of these terms in the literature to which the reviewer refers, and more explicitly define what we mean by the terms in our paper.

Questions 2-5 of Reviewer 2 concern the kinetic model that we developed. We are grateful to the Reviewer for pointing out several important issues with this part of our manuscript. In response to these comments, we revised the model, and extended the discussion clarifying all issues raised by Reviewer 2. We also revised Fig. 5 and Fig. 6 of our manuscript. We now give point-by-point response to the comments. 

Q2. An often-mentioned topic by the authors/papers mentioned in 1 is that when it comes to

“multivalent” binding, statistical factors need to be taken into account. As discussed also in details in

Chem Soc Rev 2011, 40, 1305, the 2 to 1 interaction is often used as an archetypal example.

Accordingly, care needs to be taken when discussing a 2:1 (or 1:2 – the difference really is only about

which partner in the interaction is defined as the first one) interactions, do the authors mean, e.g.,

the microscopic (Km) or macroscopic (K) binding constants, or do they mean the overall binding

constants (which in chemistry is sometimes shown as β to avoid confusion) in their description of

say, KI in Figure 5/ eq 4 which is shown there in the power of 2: KI2 ? By the looks of it, KI is really the inverse microscopic binding constant (here it does not help that chemist usually use association

constants while biologists prefer dissociation constants which are simply the inverse of the former)

for each stepwise interactions and, additionally, the authors have made the assumptions to use the

language from Chem Soc Rev 2011, 40, 1305 that KI = K1m = K2m, i.e. the stepwise interactions have

no cooperativity. It does not help here that in Eq 2, KI appears to be what in chemistry would be

called the overall binding constant with the units M-2.

Q3. Continuing with the equations: Fig 5 / eq 4 has KI

2 = ki’/ki in which the two rate constants come

from eq 3. But this does not make a lot of sense? This must be a stepwise process (here <->

equilibrium arrows:

E + I <-> EI step 1, rate constants are k1 and k-1

EI + E <-> EI step 2, rate constants are k2 and k-2

Hence Eq 4 seems incorrect. This is also evident if one considers that for Eq 4 to be correct, ki’/ki

must have the units M2.

A2 and A3. We now respond to comments (2) and (3). The Reviewer was entirely correct in bringing the issue of a two-step reaction mechanism. We thus revised the model, incorporating the two-step mechanism suggested by Reviewer in Q3. This is reflected in our revised manuscript in section ‘Kinetic model for APPI homodimer binding to mesotrypsin’ (see p. 15-20, the changes are marked in red). We also provided the entire derivation of all theoretical results without omitting any steps. We explicitly, carefully defined all parameters that we used, including all equilibrium constants appearing in the model. We believe therefore that we addressed important issues regarding the definition and the clarity of presentation raised by Reviewer in Q2. All issues regarding the definitions of the equilibrium constants were resolved by explicitly defining all these equilibrium constants and all reaction rates for both steps in the reaction mechanism. We also added an extensive discussion about the mechanism of the observed cooperativity (see the discussion on p. 18-19, and Conclusions, p. 22 in the revised manuscript).

Reviewer writes: Hence Eq 4 seems incorrect.

Regarding to Eq. 4 of our original manuscript – we clarified this issue (see the discussion after Eq. 10, Eq. 11, and Eq. 12, p. 16 in the revised manuscript). 

Q4. Continuing to Eq 5, I cannot see how it was obtained from Eq 3. Whether my concerns about Eq 3

above are correct or not, the authors need to give more details on this derivation, at least in SI (write

it out in full if necessary).

A4. We clarified this issue including all steps of the derivation (see Eq. 15-20 and related explanation in the text of the revised manuscript). 

Q5. Figure 6 – which non-linear models / equations? This is really not clear at all. Again, more details,

and ideally, raw data file etc should be included as SI.

A5. We recomputed Fig. 6 based on the revised model and clarified the caption. The title of Fig. 6 is now ‘Two-step model (Fig. 5) for mesotrypsin inhibition by APPI homodimers provides an excellent fit to experimental data’. In addition, a raw data file is now added as SI. 

Q6. Eq 8 on page 22 and related discussion earlier in the paper: Why does the bivalent inhibitor not

bind to two trypsin (but only with mesotrypsin)? Or is this not correctly understood.

A6. We thank the reviewer for this comment. Our data support the model that the dimeric APPI binds simultaneously two units of trypsin during the active site titration experiment (see Table S1). We have attempted to clarify this point through altered presentation of Table S1 and edits to the text (page 11). Because of the structural similarity of trypsin family members, we anticipate that the APPI dimer is also capable of binding simultaneously to two molecules of mesotrypsin. 

Q7. Eq 9 seems to be the main method to measure inhibition. But isn’t this very risky? As the

Graphpad manual points out this should not be used for tight binding, rather, using the Morrison

equation:

GraphPad Prism 9 Curve Fitting Guide - Equation: Tight inhibition (Morrison equation)

Now, a companying paper Microsoft PowerPoint - 1305-arqule-002.ppt (biokin.com) explains this

issue in more details.

But on reading about the Morrison Equation, it seems it is a quadratic question that is strongly

related to those used to find free ligand concentration in a typical 1:1 equilibria. If that is so, there

should be a cubic equation equivalent of the Morrison equation that would be related to the cubic

one for 1:2 or 2:1 equilibria in Chem Soc Rev 2011, 40, 1305?

A7. We thank the reviewer for this comment. We used competitive inhibition as our initial approach for data analysis for consistent comparison with prior mesotrypsin-APPI studies. We have substantial experience using this approach and also using the Morrison equation and alternative kinetic treatments appropriate for measuring tight-binding inhibition, in the context of evaluating trypsin and mesotrypsin inhibitors. We have found that the competitive inhibition approach has good sensitivity to discriminate among equilibrium inhibition constants down to the very low nanomolar range, particularly when using an appropriate substrate concentration range sampling concentrations both above and below Ki. Alternative tight-binding approaches are needed for accurate discrimination between Ki values in the sub-nanomolar range. We have found tight-binding and competitive inhibition experimental designs to give similar results for inhibitors with Ki in the 1-5 nM range. As our fits of competitive inhibition studies yielded Ki values in the mid-nanomolar range, and we were able to measure rates accurately when using substrate concentrations well below Ki, we did not use tight-binding models in this case.

Q8. Somewhat related to 7 is that the authors say on page 23 that there is excess of the inhibitor that

argue therefore I think that the “tight binding” models might not be necessary. But based on

experience from supramolecular chemistry, even with such excess this is a dangerous assumption to

make at any point for 1:2 and 2:1 equilibria.

A8. Our enzyme concentration is 0.25 nM in the mesotrypsin assays, and thus inhibitor concentrations are well in excess of enzyme in these experiments. As noted above, we have consistently used this assay and data fitting approach to reproducibly differentiate the equilibrium inhibition constants for inhibitors down to 1-2 nM, more than an order of magnitude below any Ki values measured in the present study.

Q9. And on the topic of concentration – is the concentration of the dimer inhibitor reported in various

plots etc, the concentration of the molecule or the concentration of binding loops? The latter would

be 2x the former. I mentioned this because IF the assumption is made that there is no cooperativity

between the first and second enzyme binding to the inhibitor, then it should be possible to model

the system as a simple 1:1 system provides the 2x inhibitor concentration is used in all the data

analysis.

A9. We thank the reviewer for this comment. We report the concentration of the molecule where we validate of inhibitor integrity/activity by comparing the concentrations determined by absorbance at 280 nm with those determined by bovine trypsin titration using different stoichiometry models. By contrast, in our initial analysis of Ki vs. mesotrypsin, we used the active site concentration, i.e. the concentration of binding loops (2× the concentration of the molecule). In this case, we are using the classic competitive inhibition model for a simple 1:1 system under the assumption that there is no cooperativity between the first and second enzyme binding to the inhibitor, as suggested by the reviewer. We subsequently consider a more complex treatment that does not make this assumption, as described below and in the text. We have edited the text to better clarify these points (page 14).

Q10. It would also help if the authors simply included as much as possible of their raw data and fitting

program(s) as an attached (could be deposited on github or figshare).

A10. As requested by the reviewer, a raw data file is now added as SI. The procedure of fitting the raw data using Eq. (18) is straightforward and it simply requires plotting Eq. (18) at different values of ��I. This was done using a MATLAB script. We specified all the parameters in the caption of Fig. 6. 

Q11. In conclusions, this paper appears to be an important contribution to understand avidity /

multivalency / cooperativity better. But a quite a number of issues need to be cleared up first and

the data analysis explained in enough details to make it both reproducible and unambiguous in

terms of links to other related work, particularly in chemistry.

A11. We thank the reviewer for careful reading and many helpful suggestions, and believe that our responsive revisions have greatly improved the manuscript.

---

## [Decision Letter · Decision Letter 1]

18 Oct 2021

PONE-D-21-08844R1Avidity observed between a bivalent inhibitor and an enzyme monomer with a single active sitePLOS ONE

Dear Dr. Papo,

Thank you for submitting your manuscript to PLOS ONE. After careful consideration, we feel that it has merit but does not fully meet PLOS ONE’s publication criteria as it currently stands. Therefore, we invite you to submit a revised version of the manuscript that addresses the points raised during the review process.

In your revised version please address the minor issue raised by Reviewer 2. Please ensure that your decision is justified on PLOS ONE’s publication criteria and not, for example, on novelty or perceived impact.

We look forward to receiving your revised manuscript.

Kind regards,

Israel Silman

Academic Editor

PLOS ONE

Journal Requirements:

Reviewers' comments:

Reviewer's Responses to Questions

**Comments to the Author**

1. If the authors have adequately addressed your comments raised in a previous round of review and you feel that this manuscript is now acceptable for publication, you may indicate that here to bypass the “Comments to the Author” section, enter your conflict of interest statement in the “Confidential to Editor” section, and submit your "Accept" recommendation.

Reviewer #1: All comments have been addressed

Reviewer #2: (No Response)

2. Is the manuscript technically sound, and do the data support the conclusions?

Reviewer #1: Yes

Reviewer #2: Yes

3. Has the statistical analysis been performed appropriately and rigorously? 

Reviewer #1: Yes

Reviewer #2: Yes

4. Have the authors made all data underlying the findings in their manuscript fully available?

Reviewer #1: Yes

Reviewer #2: Yes

5. Is the manuscript presented in an intelligible fashion and written in standard English?

Reviewer #1: Yes

Reviewer #2: Yes

6. Review Comments to the Author

Reviewer #1: (No Response)

Reviewer #2: I thank the authors for the revision of their manuscript and for having by and large addressed all my concerns and questions. I have only one remaining suggestion, and the editor does not need to send the paper back to me should the authors do this:

The derivation of what is equation 15 is still not clear. Yes, it is obtained by combining Eq 3, 6 and 10. But exactly how that leads to 15 is a bit unclear. Perhaps include a more detailed step-by-step derivation in the SI?

7. PLOS authors have the option to publish the peer review history of their article (what does this mean?). If published, this will include your full peer review and any attached files.

Reviewer #1: No

Reviewer #2: No

---

## [Author Response · Author response to Decision Letter 1]

11 Nov 2021

Reviewer #2: I thank the authors for the revision of their manuscript and for having by and large addressed all my concerns and questions. I have only one remaining suggestion, and the editor does not need to send the paper back to me should the authors do this:

The derivation of what is equation 15 is still not clear. Yes, it is obtained by combining Eq 3, 6 and 10. But exactly how that leads to 15 is a bit unclear. Perhaps include a more detailed step-by-step derivation in the SI?

Answer: We thank the reviewer for the comment and added a more detailed step-by-step derivation in the SI section.

---

## [Editor Report · Decision Letter 2]

15 Nov 2021

Avidity observed between a bivalent inhibitor and an enzyme monomer with a single active site

PONE-D-21-08844R2

Dear Dr. Papo,

We’re pleased to inform you that your manuscript has been judged scientifically suitable for publication and will be formally accepted for publication once it meets all outstanding technical requirements.

Kind regards,

Israel Silman

Academic Editor

PLOS ONE
---

## [Editor Report · Acceptance letter]

16 Nov 2021

PONE-D-21-08844R2 

Avidity observed between a bivalent inhibitor and an enzyme monomer with a single active site 

Dear Dr. Papo:

I'm pleased to inform you that your manuscript has been deemed suitable for publication in PLOS ONE. Congratulations! Your manuscript is now with our production department. 

Kind regards, 

on behalf of

Prof. Israel Silman 

Academic Editor

PLOS ONE